# Clustering then Propagation: Select Better Anchors for Knowledge Graph Embedding

**Ke Liang**[1]  **Yue Liu**[1]  **Hao Li**[1]  **Lingyuan Meng**[1]  **Suyuan Liu**[1]
**Siwei Wang**[2]  **Sihang Zhou**[1]  **Xinwang Liu**[1*]
[1]National University of Defense Technology, Changsha, China
[2]Academy of Military Sciences, Beijing, China

## Abstract

Traditional knowledge graph embedding (KGE) models map entities and relations to unique embedding vectors in a shallow lookup manner. As the scale of data becomes larger, this manner will raise unaffordable computational costs. Anchor-based strategies have been treated as effective ways to alleviate such efficiency problems by propagation on representative entities instead of the whole graph. However, most existing anchor-based KGE models select the anchors in a primitive manner, which limits their performance. To this end, we propose a novel anchor-based strategy for KGE, i.e., a relational clustering-based anchor selection strategy (RecPiece), where two characteristics are leveraged, i.e., (1) representative ability of the cluster centroids and (2) descriptive ability of relation types in KGs. Specifically, we first perform clustering over features of factual triplets instead of entities, where cluster number is naturally set as number of relation types since each fact can be characterized by its relation in KGs. Then, representative triplets are selected around the clustering centroids and further mapped into corresponding anchor entities. Extensive experiments on six datasets show that RecPiece achieves higher performances but comparable or even fewer parameters compared to previous anchor-based KGE models, indicating that our model can select better anchors in a more scalable way.

## 1 Introduction

Knowledge graphs (KGs) [35], such as Freebase [5], Wikidata [69], consist of a large number of relational facts, such as Freebase [5], Wikidata [69], YAGO [51] and NELL [8], consist of a large number of relational facts, which are generally in the format of triplets, *i.e., (head entity, relation, tail entity)*. Each triplet in KGs reveals a specific connection between entities. To leverage such informative knowledge to enhance the capacity of models in different fields [75, 24, 84] and applications [40, 43, 38], multiple knowledge graph embedding (KGE) models [59, 60, 48, 42, 82, 32, 31, 15, 14, 34, 35, 36, 81, 80] have been proposed these years.

However, traditional KGE models usually encode the entities, relations, and factual triplets in KGs shallowly. Assuming the dimension as $d$, traditional KGE models, such as RotatE [61], will map these elements in KGs into subspace $\mathbb{R}^{N \times d}$, where $N$ is the number of the target objects. Such a shallow lookup manner in these traditional KGE models results in a linear growth of memory consumption for storing the embedding matrix and incurs high computational costs [13]. Thus, as the scale of data becomes larger and larger, top-level GPU or CPU clusters with more memory space are required for these traditional KGE models. For example, about 78M $\times$ 200$d$ entity feature matrix and 58.1 GB GPU RAM [20] are needed for the best-performing model on PyTorch-BigGraph dataset [29].

---

*Corresponding author

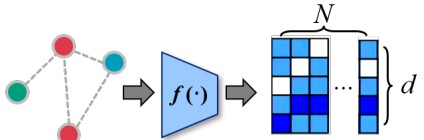
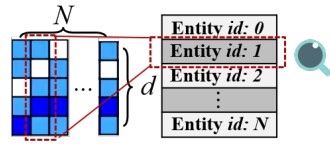

Shallow Knowledge Graph Embedding    Complexity Increases along with Entity Space

Figure 1: The problem of shallow knowledge graph embedding method.

To address such efficiency problems, there are three typical strategies integrated with KGE models, including quantification [37, 53, 71], knowledge distillation [22, 72, 55, 85], and anchor-based sampling [20, 70, 30, 78]. However, the first two types of strategies aim to realize more lightweight models for deployment by compressing the normal-size trained model. In other words, the standard KGE models still need to be trained on large datasets in advance. Compared to them, the anchor-based sampling strategy, raised by [20], will be more efficient in both training and deployment phrases, since it actually reduces the size of the entity set for propagation, *i.e.,* from an intact set of all of the entities to an anchor set of some representative entities. Note that propagation means the feature aggregation procedure. Compared to quantification and knowledge distillation, anchor-based sampling can be easily composed with different KGE baselines to optimize the efficiency of these models.

While, most of the existing anchor-based KGE models [20, 70, 30, 78] select the anchors in a primitive manner, *e.g.,* random selection, and manual selection of different centrality measurement strategies [20], etc. Thus, the anchor quality cannot be well guaranteed, which limits the performance of the models. In particular, the weights for each strategy on different datasets are usually determined according to grid searching, which is resource-consuming. In addition, considering efficiency will generally bring performance loss compared to the selected shallow KGE backbones. Therefore, reducing performance loss while ensuring good efficiency is also a problem that our work expects to solve. More related works are discussed in Appendix A.2 due to the space limitation.

Our work takes an attempt to design a more reasonable and accurate anchor selection strategy for better-quality knowledge embedding. During the investigation, two important **characteristics** come to our sights and are used in our RecPiece, including: **(1)** representative ability of the cluster centroids and **(2)** descriptive ability of relation types in KGs. Specifically, cluster centroids are proven as the most representative samples within corresponding clusters in various works [67, 52, 26, 46]. Meanwhile, clustering will not introduce too many procedures via unsupervised learning techniques, which is proper to be adopted as the core mechanism for anchor selection in an efficient KGE model. Furthermore, typical clustering algorithms require two inputs, *i.e.,* clustering features and cluster number. Both of them should be determined according to the characteristics of KGs, thus leading the clustering-based mechanism more suitable for the data type of KGs. As known to all, KGs focus more on relationships between entities compared to other graph types, so each factual triplet in KGs reveals relational knowledge. In addition, triplets can be easily categorized into different clusters according to relation types in KGs. For example, *(Mike, father of, Tom)* and *(John, father of, James)* can both be characterized into same type, *i.e.,* facts to reveal the "*father of*" relationship. The characteristic shows the descriptive ability of relation types in KGs. Inspired by it, we select features of factual triplets instead of entities as the clustering features, and the number of relation types is set as the cluster number. Note both of the information can be easily obtained as the attributes in any KGs.

To this end, we propose RecPiece, a novel anchor-based KGE model with a relational clustering-based anchor selection strategy. Specifically, we perform clustering over the features of the relational facts instead of entities, where the cluster number is naturally set to the number of relation types since each fact can be characterized by its relation in KGs. Then, the representative triplets are selected around the clustering centroids, which are further mapped into corresponding anchor entities. Extensive experiments are conducted on both link prediction and entity classification among RecPiece, shallow KGE models, and typical anchor-based KGE baseline, *i.e.,* NodePiece, to demonstrate the promising capacity of our RecPiece from six aspects, *i.e.,* superiority, effectiveness, scalability, efficiency, transferability, and sensitivity. In summary, the contributions are shown from three aspects below:

- **Problem.** We analyze the limitations of previous anchor-based KGE models, and point out two useful characteristics: (1) representative ability of the cluster centroids and (2) descriptive ability of relation types in KGs, which guide RecPiece to address the limitations.

- **Method.** We design a novel anchor-based KGE model with a relational clustering-based anchor selection strategy. In particular, we perform clustering on the features of factual triplet into $|\mathcal{R}|$ (the number of relation types) clusters, which can be easily determined as the attributes of any given KGs, thus leading to a more scalable and explainable efficient KGE model.

- **Experiment.** Extensive experiments show that RecPiece can endow shallow KGE models to have better efficiency but without significant performance losses compared to other anchor-based KGE models on various downstream tasks, indicating that our model can select better anchors in a more scalable way. In particular, RecPiece is not only 84x and 1.2x lightweight than shallow KGE baseline, *i.e.,* AutoSF, and anchor-based KGE baseline, *i.e.,* NodePiece, but also makes 9.5% and 4.9% ranking performance improvements on MRR.

## 2 Related Work

This section summarizes the recent related works from three aspects: *i.e.,* traditional knowledge graph embedding (KGE) model, parameter-efficient model, and anchor-based strategy. Due to the space limitation, please refer to Appendix A.2 for details.

## 3 Method

The methodology of our RecPiece is illustrated in this section. More concretely, we first formulate the task and present the overall framework of RecPiece. Then, we further introduce the modules and procedures within RecPiece in detail, especially for the relational clustering-based anchor selection procedure. At last, we provide a comprehensive discussion on the excellent attributes of RecPiece, which is enlightening for understanding our model. The framework of RecPiece is shown in Fig. 2.

### 3.1 Prelinmary

**Task Formulation** The knowledge graph is the directed relational graph, denoted as $KG = (\mathcal{E}, \mathcal{R}, \mathcal{G})$, where $\mathcal{E}$, $\mathcal{R}$ and $\mathcal{G}$ represent the set of entities (*i.e.,* nodes), relations (*i.e.,* edge types) and fact triplets (*i.e.,* edges), respectively. Similar to typical anchor-based baseline [20], RecPiece are more like a plug-and-play auxiliary module, which can be easily applied to any KGE model to reduce the space complexity of the adopted KGE backbone. Moreover, RecPiece is evaluated on different downstream tasks, *i.e.,* link prediction and entity classification. Note that the focus of this work is not only on those ranking and classification metrics but also the efficiency. In other words, the main goal of our RecPiece is to achieve better performances on different tasks with fewer or comparable parameters compared to the previous anchor-based KGE models.

Table 1: Notation summary.

| Notation | Explanation |
|---|---|
| $\mathcal{E}, \mathcal{R}, \mathcal{G}$ | set of entity, relation and triplet |
| $e, r, t$ | element of entity, relation, fact |
| $\mathcal{G}_i$ | fact set for relation $r_i$ |
| $p(\cdot)$ | pretrained triplet encoder |
| $g(\cdot)$ | clustering algorithm |
| $\phi_a(\cdot)$ | candidate triplet selection mechanism |
| $\phi_b(\cdot)$ | triplet-entity mapping mechanisms |
| $f(\cdot)$ | KGE models for feature propagation |
| $|\cdot|$ | quantity number |
| H, h | feature matrix and vector |
| $\mathcal{C}$ | cluster centroid set |
| $c_i$ | $i^{th}$ cluster centroid |
| $dist(\cdot)$ | distance function |
| $\mathcal{D}_i$ | distance set |
| $\mathcal{T}^*, \mathcal{T}_i$ | candidate triplet set and subset for cluster $i$ |
| $\mathcal{A}$ | candidate anchor set |
| $\Theta$ | triplet distribution based on relations |
| $k$ | number of anchors (hyper-parameter) |

**Knowledge Graph Characteristic** Knowledge graphs (KGs) [35, 25] store the relational facts intuitively. Compared to other graph data, KG focuses more on the relationships between entities, and the knowledge is stored in the factual triplets. Moreover, considering adopting the clustering algorithm for anchor selection, the cluster number can be easily fetched according to the number of relation types. Thus, we selected the features of triplets as the clustering features in this paper.

### 3.2 Overview Framework

Our RecPiece is a novel anchor-based KGE model with a relational clustering-based anchor selection strategy, which contains five procedures as shown in Fig. 2, including (a) feature preparation, (b) clustering over features of factual triplets, (c.1) candidate triplet selection, (c.2) triplet-entity mapping and (d) feature propagation. As anchors in RecPiece are selected based on the clustering-based mechanism, we need to generate the clustering features and determine the cluster number in advance. Thus, encoder $p(\cdot)$ is adopted for triplet feature preparation for clustering in (a). Then, the generated features are clustered into $|\mathcal{R}|$ clusters via $g(\cdot)$ during (b). Later on, we construct the anchor set via two procedures, (c.1) and (c.2). Finally, feature propagation happens on the anchors constrained with different task losses in (d). More details are described as follows, and notations refer to Tab. 1.

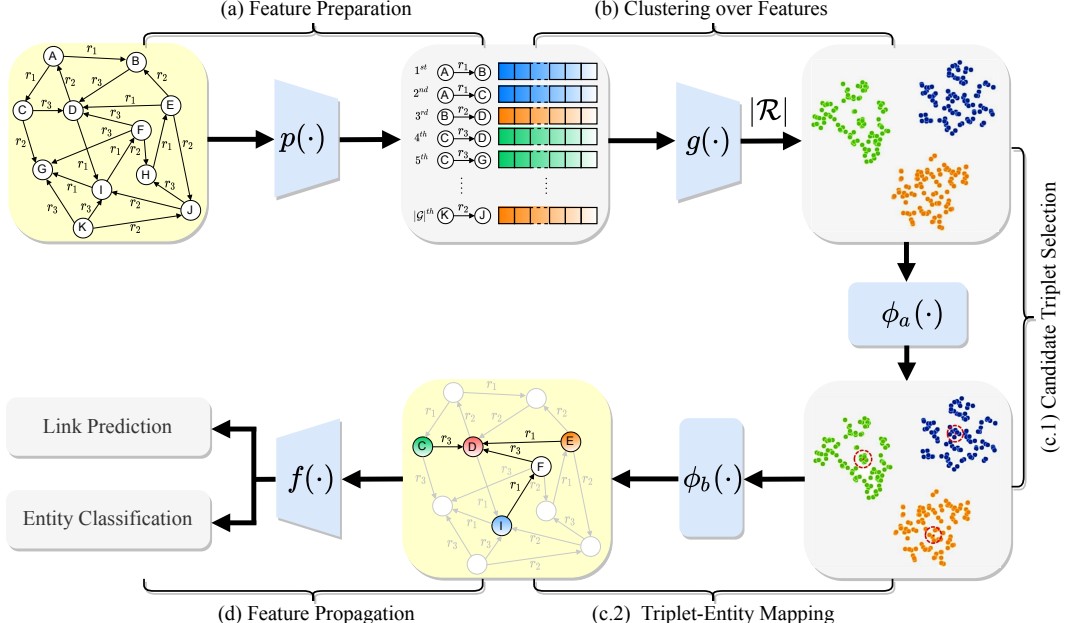

Figure 2: The illustration of our model, which is an anchor-based KGE model, termed RecPiece, by introducing a more explainable and scalable relational clustering-based anchor selection strategy. Note that different factual triplets are coloured in different colours according to relation types, *i.e.,* blue for $r_1$, orange for $r_2$, and green for $r_3$. $p(\cdot)$ and $f(\cdot)$ are two encoders for feature preparation and propagation, respectively. Besides, $g(\cdot)$ is the adopted clustering algorithm with the cluster number set as the number of relation type $|\mathcal{R}|$, and anchor set construction contains two steps $\phi_a(\cdot)$ and $\phi_b(\cdot)$. Note that the detailed description of the above modules is illustrated in Section 3 and the notations are summarized in Tab. 1.

### 3.3 Feature Preparation

Feature preparation aims to generate the features for clustering. During the procedure, the encoder $p(\cdot)$ takes the triplet set $\mathcal{G}$ as input and outputs the corresponding feature matrix $\mathbf{H}_\mathcal{G} \in \mathbb{R}^{|\mathcal{G}| \times d}$, where the feature vector $\mathbf{h}_{t_i}$ of triplet $t_i$ can be easily fetched from the corresponding row as $\mathbf{H}_\mathcal{G}[i, :]$.

$$\mathbf{H}_\mathcal{G} = p(\mathcal{G}) \tag{1}$$

Specifically, $p(\cdot)$ contains the following steps: we first generate the embeddings of entities and relations with selected knowledge graph encoders. Then, we traverse all factual triplets in $\mathcal{G}$ and get the feature vector $\mathbf{h}_{t_n}$ for $n^{th}$ triplet $t_n = (e_h, r, e_t)$ by summing up the normalized embeddings $\bar{\mathbf{h}}$ of entities ($\bar{\mathbf{h}}_{\mathbf{e_h}}$, $\bar{\mathbf{h}}_{\mathbf{e_t}}$) and relation ($\bar{\mathbf{h}}_{\mathbf{r}}$). Finally, the triplet feature matrix $\mathbf{H}_t$ is generated by concatenating all triplet embeddings together.

$$\mathbf{H}_\mathcal{G} = \bigoplus_{t_n \in \mathcal{G}} \mathbf{h}_{t_n} \tag{2}$$

### 3.4 Relational Clustering-based Anchor Selection

As the core of RecPiece, a novel relational clustering-based anchor selection strategy is designed based on characteristics of both clustering centroids and knowledge graphs. Specifically, the selection strategy can be separated into two parts, *i.e.,* cluster centroid generation and anchor set construction.

#### 3.4.1 Cluster Centroid Generation

We first perform clustering over the priorly generated triplet feature matrix, *i.e.,* $\mathbf{H}_t$ into $|\mathcal{R}|$ clusters, where cluster number is set as the number of the relation types. The adopted clustering algorithm $g(\cdot)$ can output the embedding vector set of cluster centroids $\mathcal{C} = \{\mathbf{h}_{c_1}, \mathbf{h}_{c_2}, \cdots, \mathbf{h}_{c_{|\mathcal{R}|}}\}$.

$$\mathcal{C} = g(\mathbf{H}_t, |\mathcal{R}|) \tag{3}$$

### 3.4.2 Anchor Set Construction

The cluster centroids are usually not the specific samples in datasets. Thus, we find representative samples as anchors around them. To achieve the goal, two procedures are designed for anchor set construction, including candidate triplet selection $\phi_a(\cdot)$ and triplet-entity mapping $\phi_b(\cdot)$. Concretely, the former procedure aims to select representative triplets for each relation type, while the latter procedure picks up the representative entities based on the triplets to constitute the final anchor set.

**Candidate Triplet Selection.** The factual triplets with the top $m_i$ closest distance to the cluster centroid $c_i \in \mathcal{C}$ are selected as the candidate triplets. Specifically, *cosine(·)* function is used as the distance function $\mathrm{dist}(\cdot)$ to measure the distance, which can also be substituted into other functions, such as *euclidean(·)*. Then, we can get the distance set $\mathcal{D}$ for each cluster:

$$
\begin{aligned}
\mathcal{D}_i &= \bigcup_{t_n \in \mathcal{G}} \mathrm{dist}(c_i, t_n) \\
&= \bigcup_{t_n \in \mathcal{G}} \left[ \frac{\langle \mathbf{h}_{c_i}, \mathbf{H}_t[n, :] \rangle}{\|\mathbf{h}_{c_i}\|_2 \cdot \|\mathbf{H}_t[n, :]\|_2} \right],
\end{aligned}
\tag{4}
$$

where $t_n \in \mathcal{G}$ denote the $n^{th}$ factual triplet, and $\mathbf{h}_{c_i}$ represents the feature vector for centroid $c_i$.

$$
\begin{aligned}
\mathcal{T}^* &= \bigcup_{i \in [1, |\mathcal{R}|]} \mathcal{T}_i \\
&= \bigcup_{i \in [1, |\mathcal{R}|]} \arg \mathrm{top}\text{-}m_i \; \mathcal{D}_i \,,
\end{aligned}
\tag{5}
$$

where the candidate triplet set $\mathcal{T}^*$ is composed of sets $\mathcal{T}$ for different relation types. Triplets, whose embeddings are the top $m_i$ closest to $\mathbf{h}_{c_i} \in \mathcal{C}$, are selected as the candidate triplets via $\arg \mathrm{top}\text{-}m_i$. Note that $m_i$ is not a hyperparameter, and it can be calculated when given the total anchor number $k$:

$$
m_i = k \cdot \frac{|\mathcal{G}_i|}{|\mathcal{G}|},
\tag{6}
$$

where $r_i$ represents the $i^{th}$ relation type and $|\mathcal{G}_i|/|\mathcal{G}|$ is the frequency distribution $\Theta$ of triplets with relation $r_i$, which can be easily obtained based on the attribute of the KGs. Taking the KG in Fig. 2 as an example, the $[|\mathcal{G}_1|, |\mathcal{G}_2|, |\mathcal{G}_3|] = [7, 7, 7]$ and $r_3$, and if $k = 6$, $[m_1, m_2, m_3] = [2, 2, 2]$.

**Triplet-Entity Mapping.** The entity anchor set $\mathcal{A}$ is constructed from the selected triplet set $\mathcal{T}^*$ by randomly picking either the head or tail entity as the anchor entity corresponding to each triplet. The $|\mathcal{A}| \leq |\mathcal{T}^*|$, since we will remove those identical entities.

$$
\mathcal{A} = \phi_b(\mathcal{T}^*)
\tag{7}
$$

### 3.5 Feature Propagation

The hashing and encoding procedures in [20] are leveraged for feature propagation. Specifically, the features of each entity $e_i$ are first hashed into a *hash(e_i)* using two types of anchor information, *i.e.,* discrete distances and relational contexts. Then, we leverage MLP as the encoder $f(\cdot)$ to bootstrap the feature embeddings of each entity based on the vectorized hashing features. Besides, according to the types of the downstream tasks, *i.e.,* entity classification and link prediction, various loss functions are adopted for training and optimization. In conclusion, our RecPiece can be integrated with different combinations of KGE backbones and loss functions toward different tasks and scenarios.

### 3.6 Attributes of RecPiece

In this section, we further discuss some attributes of the proposed RecPiece from various aspects shown below. (1) Random and manual anchor selection in previous anchor-based models is highly dependent on the human experience. Compared to them, ours is more reasonable and learnable according to the representative ability of the cluster centroids. (2) Our RecPiece is developed based on the characteristics of KGs. Specifically, we perform clustering on features of triplets instead of entities since the knowledge units in KG are stored in triplets, which can also be easily characterized based on the relation type. (3) The hyper-parameter, *i.e.,* cluster number, for clustering algorithms can be determined according to the attributes in KGs in RecPiece as the number of relation types. Thus, our anchor selection only contains one hyper-parameter, *i.e.,* anchor number, which is inevitable and

Table 2: Link prediction results on FB15k-237, WN18RR, CoDEx-L, and YAGO 3-10. % denotes the Hits@10 ratio regard to the anchor-based KGE baseline, *i.e.,* NodePiece + RotatE.

| | FB15k-237 | | | | | WN18RR | | | | |
|---|---|---|---|---|---|---|---|---|---|---|
| | #Parameter (M) | MRR | Hits@10 | % | *Effi.* | #Parameter | MRR | Hits@10 | % | *Effi.* |
| RotatE | 29.3 | 0.338 | 0.533 | 100 | 0.012 | 40.6 | 0.476 | 0.571 | 100 | 0.012 |
| NodePiece + RotatE | 3.2 | 0.254 | 0.420 | 78.8 | 0.079 | 5.0 | 0.396 | 0.504 | 88.3 | 0.079 |
| RecPiece + RotatE | 2.9 | 0.265 | 0.431 | 80.9 | 0.091 | 5.0 | 0.402 | 0.506 | 88.6 | 0.080 |
| Improvement | 9.3% | 4.3% | 2.6% | 2.6% | 1.5% | 0% | 0.3% | 0.4% | 0.3% | 1.3% |
| | CoDEx-L | | | | | YAGO 3-10 | | | | |
| | #Parameter (M) | MRR | Hits@10 | % | *Effi.* | #Parameter | MRR | Hits@10 | % | *Effi.* |
| RotatE (500d) | 77.0 | 0.258 | 0.387 | 100 | 0.003 | 123.0 | 0.495 | 0.670 | 100 | 0.004 |
| RotatE (20d) | 3.8 | 0.196 | 0.322 | 83.2 | 0.052 | 4.8 | 0.121 | 0.262 | 39.1 | 0.025 |
| NodePiece + RotatE | 3.6 | 0.190 | 0.313 | 80.9 | 0.053 | 4.1 | 0.231 | 0.465 | 69.4 | 0.055 |
| RecPiece + RotatE | 3.0 | 0.198 | 0.323 | 83.5 | 0.066 | 4.1 | 0.243 | 0.482 | 71.9 | 0.058 |
| Improvement | 16.7% | 4.2% | 3.2% | 3.2% | 24.5% | 0% | 5.2% | 3.7% | 3.6% | 5.5% |

Table 3: Entity classification results on two subsets in WD50k.

| | | WD50K (5% labeled) | | | WD50K (5% labeled) | | |
|---|---|---|---|---|---|---|---|
| | #Parameter (M) | ROC-AUC | PRC-AUC | Hard Acc | ROC-AUC | PRC-AUC | Hard Acc |
| MLP | 4.1 | 0.503 | 0.016 | 0.001 | 0.510 | 0.017 | 0.002 |
| COMPGCN | 4.4 | 0.836 | 0.280 | 0.176 | 0.834 | 0.265 | 0.161 |
| Nodepiece+COMPGCN | 0.75 | 0.981 | 0.443 | 0.513 | 0.981 | 0.450 | 0.516 |
| RecPiece + COMPGCN | 0.64 | 0.983 | 0.459 | 0.538 | 0.984 | 0.464 | 0.536 |
| Improvement | 14.7% | 0.3% | 3.6% | 4.9% | 0.3% | 3.1% | 3.9% |

also needed by other anchor-based methods. Besides, other models even need resource-consuming grid-searching to get weights for different centrality measurement strategies on different KGs. (4) Based on the above analyses, RecPiece is also more capable of being extended to various KGE models and applied to different KGs.

# 4 Experiment

Experiments are conducted to demonstrate the promising capacity of our RecPiece from five aspects, *i.e.,* superiority, effectiveness, scalability, efficiency, transferability, and sensitivity, by answering the following six questions.

- **Q1: Superority.** Does RecPiece achieve better performance compared to the previous anchor-based strategy, NodePiece, when integrated with different KGE models for different downstream tasks?

- **Q2: Effectiveness.** Does the clustering strategy make a difference? Besides, how do the adopted components in RecPiece influence the performance?

- **Q3: Efficiency.** Will the RecPiece lead to a more parameter-efficient model? What is the performance of time and memory cost?

- **Q4: Scalability.** How does our RecPiece perform on large-scale knowledge graph?

- **Q5: Transferability.** Will our RecPiece be effectively integrated with different KGE backbones?

- **Q6: Sensitivity.** How does the performance influenced by RecPiece with different hyper-parameters?

## 4.1 Experiment Setting

**Datasets.** Six benchmark datasets are leveraged to evaluate our RecPiece as same as previous works do [13, 20, 30]. Specifically, FB15k-237 [64], WN18RR [18], CoDEx-L [54], and YAGO3-10 [45] are used for link prediction. The entity classification is carried out on two subsets (5% and 10% labeled) from WD50K [21], and the OGB WIKIKG 2 [23, 20] is the larger KGs for scalability analysis. The statistic details of the datasets are shown in Tab. 9.

Table 4: Dataset Statistic. "LP" and "EC" denote link prediction and entity classification. "#" represents the number.

| Data | Task | #Ent. | #Rel. | #Fact |
|---|---|---|---|---|
| FB15k-237 | LP | 14,505 | 237 | 310,079 |
| WN18RR | LP | 40,559 | 11 | 92,583 |
| CoDEx-L | LP | 77,951 | 69 | 612,437 |
| YAGO3-10 | LP | 123,143 | 37 | 1,089,000 |
| OGB WikiKG 2 | LP | 2,500,604 | 535 | 17,137,181 |
| WD50K | EC | 46,164 | 526 | 222,563 |

**Implementation Details.** All experiments are conducted on the server with 4-core Intel(R) Xeon(R) Platinum 8358 CPUs @ 2.60GHZ, a single 80 GB A100 GPU and 64GB RAM with PyTorch [49] libraries.

The $p(\cdot)$ for feature preparation is selected as pretrained NodePiece [63] in the first few epochs. Besides, k-means [41, 44] is selected as $g(\cdot)$ for clustering, and the cluster number is set as "#Rel." in Tab. 9 for different datasets. For a fair comparison, we set anchor numbers $k$ for each dataset as the same as [20], and 2-layer-MLP is adopted as $f(\cdot)$ feature propagation. In addition, we replace the default $p(\cdot)$, $g(\cdot)$ and $dist(\cdot)$ to pretrained GraIL [63], BitechingK-means [50] and euclidean($\cdot$) for robustness analysis. Moreover, as for different tasks, RecPiece is integrated with three KGE backbones, *i.e.,* Ro-

Table 5: Ablation study for different anchor selection strategies."EP" and "RP" represent entity prediction and relation prediction. "NDC" and "PPR" are short for Node Degree Centrality and Personalized PageRank.

| | Entity Prediction | | Relation Prediction | |
|---|---|---|---|---|
| | MRR | Hits@10 | MRR | Hits@10 |
| Random | 0.249 | 0.417 | 0.878 | 0.971 |
| NDC | 0.250 | 0.418 | 0.877 | 0.970 |
| PPR | 0.251 | 0.419 | 0.878 | 0.971 |
| NodePiece | 0.254 | 0.420 | 0.881 | 0.970 |
| RecPiece | 0.265 | 0.431 | 0.884 | 0.975 |

tatE [61], ComPGCN [68], and AutoSF [79] to compare with thirteen KGE models, including (1) link prediction: TransE [6], DisMult [77], ComplEX [65], PairRE [10], RotatE [61], TripleRE [78], AutoSF [79], LRE + PairRE [11], NodePiece + RotatE [20], and NodePiece + AutoSF [20]; (2) entity classification: MLP, ComPGCN [68], and NodePiece + ComPGCN [20]. More details are present in Appendix.

**Evaluation Metrics.** For link prediction, both MRR [16] and Hits@k [1] are used as the ranking metrics. Besides, ROC-AUC, PRC-AUC, AP, and Hard Accuracy are the evaluation metrics [23] for entity classification. To quantify the efficiency, we report the parameter number #P (M), memory cost (GB), running time (hours), and *Effi.* [13]. Note that *Effi.* [13] is calculated by MRR/#P.

## 4.2 Main Performance (RQ1)

The performance comparison is carried out between our RecPiece and the existing anchor-based KGE baseline, *i.e.,* NodePiece, on two typical downstream tasks, *i.e.,* link prediction and entity classification. It aims to answer **Q1**.

**Results Report** Tab. 2 shows that RecPiece can achieve better performance on link prediction, *i.e.,* average 6.5% fewer on parameter number and 3.5% and 2.5% improvements on MRR and Hits@10. In particular, the improvements are apparent on FB15k-237 and CoDEx-L. Even though the performance improvements on WN18RR are the smallest, it is still comparable. According to Tab. 3, RecPiece can also achieve promising results on entity classification with about 14.7% fewer in parameter number and 3.3% and 4.4% boosts on PRC-AUC and Hard ACC, respectively.

Table 6: Ablation study for whether pretrained based on language models. "KG-self" and "PLM" represent that the pretrained features are generated on structure information and extra-textual information, respectively.

| Model | MRR | Hits@10 |
|---|---|---|
| RotatE | 0.338 | 0.533 |
| NodePiece + RotatE | 0.254 | 0.420 |
| RecPiece (KG-self) + RotatE | 0.265 | 0.431 |
| RecPiece (PLM) + RotatE | 0.262 | 0.425 |

Table 7: Ablation study for different clustering features. "triplet" and "entity" represent clustering over the features of relational triplets and entities, respectively. All the results are for link prediction results on FB15k-237

| Model | MRR | Hits@10 |
|---|---|---|
| RotatE | 0.338 | 0.533 |
| NodePiece + RotatE | 0.254 | 0.420 |
| RecPiece (triplet) + RotatE | 0.265 | 0.431 |
| RecPiece (entity) + RotatE | 0.259 | 0.424 |

**Discussion** Based on the above results, we can easily get the answer to **Q1** that our RecPiece can achieve better performances on both link prediction and entity classification with comparable or even fewer parameters compared to the previous anchor-based strategy, NodePiece [20]. It further indicates less performance loss will be caused by RecPiece in a more parameter-efficient manner. Although our RecPiece still raises the performance loss compared to the shallow KGE baselines, *i.e.,* RotatE and COMPGCN, it is an inevitable trade-off for considering efficiency (over 90% reduction on parameters) via anchor-based strategy, which also occurs on other anchor-based KGE models. Moreover, we notice that different performance improvements are made by our RecPiece in different datasets. It may suggest that our RecPiece can achieve better performance on those denser datasets for link prediction, thus leading to fewer improvements on sparser WN18RR compared to denser FB15k-237.

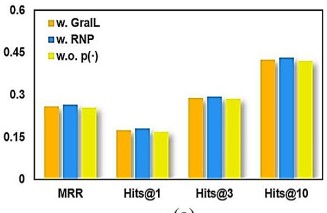 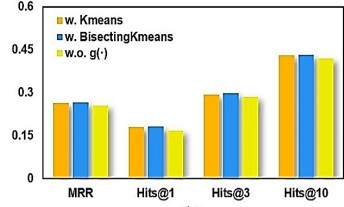 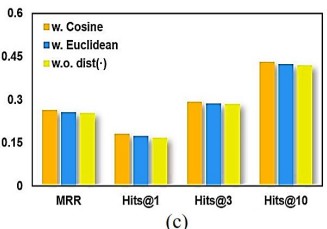

Figure 3: Ablation study of different components in RecPiece. (a), (b) and (c) show the impact of the different pretrained triplet encoder $p(\cdot)$, clustering algorithm $g(\cdot)$, and distance function $\mathrm{dist}(\cdot)$ for link prediction task on FB15k-237

## 4.3 Ablation Studies (RQ2)

In this section, different ablation studies are presented to prove the effectiveness of RecPiece. Concretely, we first discuss the effectiveness of our anchor-selection strategy. Then, we further analyze the effectiveness of the important components in each step of RecPiece shown in Section 3.2 and Fig. 2.

### 4.3.1 Anchor Selection Strategies

The anchors of better quality will definitely contribute to better performances. To prove that our relational clustering-based anchor selection strategy can effectively select better anchors, we compare it with four other strategies, *i.e.,* Random Selection, Node Degree Centrality, Personalized PageRank, and NodePiece, on FB15k-237 for link prediction, which contains two different settings, *i.e.,* missing entity prediction and missing relation prediction. Tab. 5 shows that our anchor selection strategy outperforms other strategies in both two settings. In particular, our strategy makes the 4.3% and 2.6% performance boost on MRR and Hits@10 metrics compared to the NodePiece-based model.

### 4.3.2 Feature Preparation

In this section, we discuss the impact of different pretrain triplet encoders in the first step of RecPiece, *i.e.,* feature preparation.

We first adopt different structural information encoders $p(\cdot)$ in KGs for pretraining. Fig. 3 (a) shows that both GraIL [63] and default RNP [20] can both achieve promising performance, but RNP is better than the GraIL model. Besides, we also attempt to leverage the typical pretrained language model (PLM), *i.e.,* BERT [27], to prepare the pretrained features over the real textual meaning of different entities and relations. The results are shown in Tab. VII. It indicates that relying on structural information for pretraining on KG link prediction is more promising than extra-textual meaning. It is reasonable that link prediction is indeed a task more related to network structures. However, the results also show the potential capacity of our RecPiece when leveraging the extra information.

In our model, the pretrained procedure is only used for feature preparation, which can be replaced as you want. No matter which feature preparation it is, the key idea of the paper will not be affected. Nevertheless, our RecPiece can all make improvements when leveraging different pretrained encoders.

### 4.3.3 Clustering over Features

As for the second step of RecPiece, we discuss and analyze the clustering features and the clustering methods.

**Clustering Features**  We also conduct experiments to verify that it is better to perform clustering on relational triplet features. Table VI shows that although there are still performance boosts when leveraging entity features, there are more apparent improvements in performances with triplet features. Thus, the results prove our idea, *i.e.,* relational triplet features are more representative than entity features in KGs. Note that we try different cluster numbers for experiments on entity clustering and select the best results of them (with cluster number 10).

**Clustering Methods**  We also try different clustering methods for anchor selection, *i.e.,* KMeans and BisectingKmeans. Fig. 3 (b) shows that both clustering methods can lead to promising performances. It further indicates that our framework is effective with different clustering method choices, which demonstrates the generalizability of our model.

### 4.3.4 Anchor Selection

As for the anchor selection step, we further analyze the impact of different distance functions to select the anchors that are closer to the clustering centroids. The experiments are carried out on two

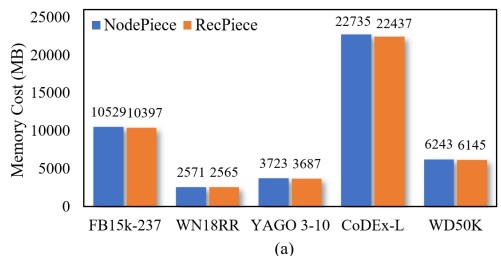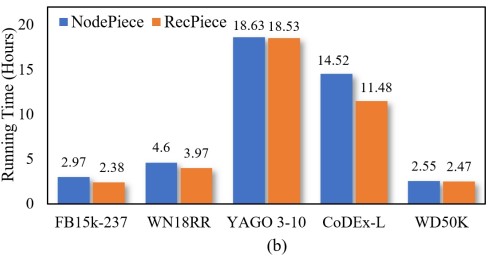

Figure 4: Memory cost and running time comparison.

types of distance function, *i.e.,* Cosine function and Euclidean function. Fig. 3 (c) demonstrates that RecPiece is robust to different distance functions, where Cosine(·) is better than Euclidean(·). But no matter which strategies they are, the performances are higher than the performance without distance function. It further indicates the effectiveness and generalizability of our framework.

### 4.3.5 Discussion

We comprehensively present ablation studies from different aspects. Tracking all of the results of the experiments demonstrates the effectiveness of RecPiece, which is composed of the answer to **Q2**. Since the ablation study on feature propagation is to evaluate the effectiveness of different combinations of backbone KGE models, which is similar to transferability analysis, the detailed discussion of this part is shown in Section 4.6.

### 4.4 Efficiency Analysis (RQ3)

This section presents and discusses the efficiency of RecPiece from three aspects, parameter efficiency, memory efficiency, and time efficiency. As shown in Tab. 2 and Tab. 3, RecPiece can incredible save about 10x, 8x, 26x, 30x, and 7x parameters on five benchmark datasets, including FB15k-237, WN18RR, CoDEx-L, YAGO 3-10 and WD50K compared to shallow KGE baselines, *i.e.,* RotatE and COMPGCN. Compared to NodePiece, the results above prove that RecPiece can endow the KGE model to be more parameter-efficiency. Specifically, performance boosts are made on both link prediction and entity classification with comparable or even fewer parameters. It

Table 8: Link prediction results on OGB WikiKG 2. The best results are marked in Bold.

| Model | #Params | MRR |
|---|---|---|
| TransE (500d) | 1250M | 0.426 ±0.003 |
| DisMult (500d) | 1250M | 0.373 ±0.005 |
| RotatE (250d) | 1250M | 0.433 ±0.002 |
| ComplEX (250d) | 1250M | 0.503 ±0.003 |
| PairRE (200d) | 500M | 0.521 ±0.003 |
| PairRE+LRE (700d) | 505M | 0.584 ± n/a |
| TripleRE | 501M | 0.579 ±0.002 |
| AutoSF | 500M | 0.546 ±0.005 |
| NodePiece + AutoSF | 6.9M | 0.570 ±0.003 |
| RecPiece + AutoSF | 5.9M | 0.598 ±0.003 |
| Improvement | 14.5% | 2.2% |

further indicates less performance loss will be caused by RecPiece in a more efficient manner. Although we focus more on parameter efficiency, Fig. 4 shows the recorded real GPU cost and running time are comparable and slightly less. Concretely, it has better efficiency on real memory cost and running time, *i.e.,* on average 114 MB and 0.89 hours reduction compared to NodePiece. It further indicates that RecPiece also has a good effect on memory efficiency and time efficiency.

In conclusion, the efficiency of our RecPiece is promising (**RQ3**). We also admit that the pre-train paradigm and anchor-based strategies will cost some resources, both in time and memory, which will be optimized in the future. Note that such redundancy is commonly seen in other methods with similar techniques. Considering the performance gains, such a limited cost is also acceptable, which should not influence the effectiveness of RecPiece.

### 4.5 Scalability Analysis (RQ4)

We compare RecPiece with nine state-of-the-art KGE models on OGB WikiKG 2 [23] to measure its scalability to larger KGs. According to Table 8, we observe that our RecPiece + AutoSF can outperform other KGE models. Specifically, the RecPiece + AutoSF model has only 5.9 M parameters, about 84x smaller than the most efficient shallow models, *i.e.,* AutoSF. Meanwhile, it is also about 1.2x lighter than the anchor-based KGE baseline, *i.e.,* NodePiece + AutoSF [20], with the same quantity of anchors. Meanwhile, our RecPiece + AutoSF can even achieve better ranking performances compared to the KGE model with best performance, *i.e.,* about 2.2% MRR improvement compared to PairRE + LRE [70]. Moreover, the running time for NodePiece + AutoSF and RecPiece + AutoSF is recorded

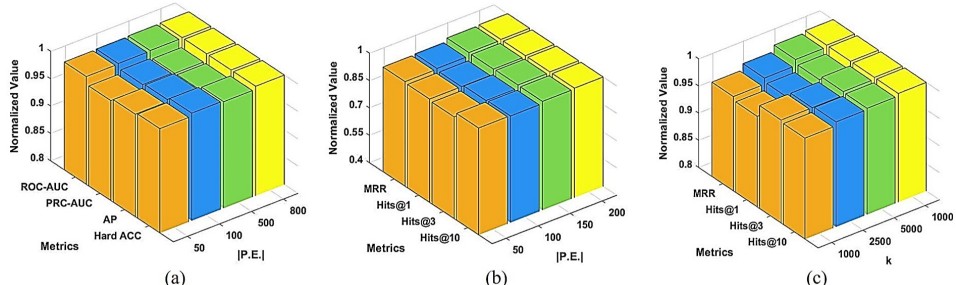

Figure 5: Sensitive analysis in different settings. (a) and (b) reveals the influence of the models on feature preparation in different epochs (*i.e.,* |P.E.| represents the pretrained epoch number), where (a) is for entity classification on WD15K (5% labeled), and (b) is for link prediction on FB15k-237. (c) indicates the influence of the total anchor number $k$. (c) is based on link prediction on FB15k-237.

as 5.33 and 5.25 hours, which is also comparable. We believe that the above results prove the better scalability of RecPiece with promising parameter reduction and better ranking performance.

### 4.6 Transferability Analysis (RQ5)

We answer **Q5** in this section. Based on the aforementioned experimental results cover two different downstream tasks, *i.e.,* entity classification and link prediction, and three different types of shallow KGE backbones, *i.e.,* RotatE, COMPGCN, and AutoSF, we can reorganize and analyze the results from another view. First of all, all of the experimental results are promising. In particular, there are 5.2% MRR performance improvements made by RecPiece on YAGO 3-10 for link prediction when integrated with RotatE. Secondly, the improvements occur in all the situations when the backbone models are integrated with our RecPiece. Therefore, as the conclusion, it is demonstrated that our RecPiece is proven to be easily extended to different tasks and different KGE models as a plug-and-play auxiliary mechanism, which shows great transferability of our RecPiece.

### 4.7 Sensitivity Analysis (RQ6)

We measure the sensitivity of RecPiece from two aspects to answer **Q6**, including the analysis on (1) pre-trained epochs for the feature preparation model $p(\cdot)$, (2) anchor number $k$. The experiment results are shown in Fig. 5. In general, our RecPiece is insensitive to the hyperparameters, which demonstrates that our RecPiece can achieve stable performances. More specifically, we can get the following two observations according to the aforementioned two aspects.

(1) Fig. 5 (a) and (b) reveals that there usually exists a pretrained epoch threshold $pe^*$ for $p(\cdot)$, which indicates that the pretrained features are effective enough to be clustered for anchor selection after $pe^*$, *e.g.,* around 800 epochs (in 4000 epochs) for entity classification on WD15K (5% labeled) and 150 epochs (in 400 epochs) for link prediction on FB15k-237.

(2) Fig. 5 (b) shows that more anchors will benefit the performance as same as NodePiece. Traditional KGE models propagate the features based on the whole graph, which is equivalent to the anchor set composed of all entities. A larger set of anchors is closer to the complete entity set so that less information will be abandoned, thus leading to better performance.

## 5 Conclusion

In this paper, we propose a novel anchor-based KGE model with a relational clustering-based anchor selection strategy, RecPiece, where two characteristics are leveraged, *i.e.,* (1) representative ability of the cluster centroids and (2) descriptive ability of relation types in KGs. Specifically, we perform clustering over the features of triplets instead of entities into $|\mathcal{R}|$ (number of relation types) clusters. Then, representative samples are selected around cluster centroids, which are further mapped into corresponding anchor entities. Extensive experiments show that RecPiece can endow shallow KGE models to have fewer parameters without significant performance loss compared to other models, on various tasks, indicating that our model selects better anchors in a more scalable way. In the future, we plan to optimize this preparation procedure via a self-adaptive mechanism along with feature propagation for better practicability and adaptivity in the future.

## Acknowledgement

This work was supported by the National Natural Science Foundation of China (project no. 62325604, 62276271).

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

# A Appendix

## A.1 Limitations and Future Works

Although proven effective and the idea is novel for large-scale KGE models, this paper only takes a primitive attempt to select better anchors via clustering. Specifically, our feature preparation is independent which requires auxiliary models. In this manner, the quality of anchors will be influenced if the adopted model is not well-trained. Thus, we plan to optimize this preparation procedure via a self-adaptive mechanism along with feature propagation for better practicability and adaptivity in the future.

## A.2 Related Work

This section summarizes the recent related works from three aspects: *i.e.,* traditional knowledge graph embedding (KGE) model, parameter-efficient model, and anchor-based strategy. The discussion on typical related works provides the necessary background knowledge for our novel anchor-based parameter-efficient KGE model, which assists readers to better understand our work.

**Traditional Knowledge Graph Embedding Model.** Traditional knowledge graph embedding (KGE) models aim to map entities and relations into low-dimensional representations [25, 2, 33] while preserving the semantic information underlying the original KG. In general, it can be roughly divided into three types, *i.e.,* translational models [6, 62, 19], tensor factorization models [57, 66], and graph neural network based models [28, 57, 17, 68, 76]. For example, RotatE [61] proposes a rotation-based translational method with complex-valued embeddings to better infer the symmetry, anti-symmetry, inversion, and composition facts, which are widely used. As an improved model for RGCN [56], COMPGCN [68] also jointly learns the representations with various entity-relation composition operations. Moreover, AutoSF [79] proposes an adaptive manner for KGE scoring function searching, which gains promising performance on large-scale scenarios. The above three typical KGE baselines constitute the backbone models in this paper following previous works [20, 10]. However, few of these traditional KGE models do consider the efficiency of parameters [20] and data compression [83]. As the scale of data becomes larger, the shallow lookup in traditional KGE models will result in linear growth of memory consumption for storing larger embedding matrices and incurs high computational costs [20], which limits their scalability to real-world scenarios. Our RecPiece is designed for more parameter-efficient representation learning on knowledge graphs.

**Parameter-Efficient Model.** Different strategies have been leveraged for more efficient KGE models in recent years, which can be roughly categorized into three types, *i.e.,* quantification [37, 53], knowledge distillation [22, 72, 85], and anchor-based sampling [39, 3]. As for quantification-based methods, TS-CL [53] reduces the dimensions by learning discrete entity representations via quantization. Besides, LightKG [71] designs the residual module to induce diversity among codebooks and performs the dynamic negative sampling using quantization technology. Compared to them, MulDE [72], DualDE [85], and Graph2Feat [55] are all distillation-based methods, which are developed based on the teacher-student frameworks. Specifically, MulDE [72] leverages multiple KGE models as teacher models to extract a student model with a lower dimension for space consumption reduction. However, the teacher model in DualDE [85] is more suitable for the student model to obtain better distillation results by considering the dual influence between them. Besides, Graph2Feat [55] extends the distillation procedure for inductive setting. **However**, the above two types of strategies aim to realize more lightweight models for deployment by compressing the trained model in normal size, which means that the standard KGE models still need to be trained on large datasets in advance. Compared to them, the anchor-based models [20, 70, 30, 78] will be more efficient in both training and deployment since they actually reduce the size of the entity set for propagation. Moreover, anchor-based sampling can be easily composed with different KGE baselines to optimize the efficiency of these models. Beyond the above categories, ATTH [9] utilizes hyperbolic space and geometric transformations to learn the improved low-dimensional representations, and PIE [12] reduces the space consumption for entity representations via discrete code vectors and tensor decomposition. More recently, EARL [13] does not learn one vector for each entity but by learning only the embedding of a small number of entities, encoding the distinguishing information from their connected relationships, k nearest reserved entities, and multi-hop neighbors, which translates the distinguishing information into entity embedding. A*Net [86] proposes a path-based propagation strategies. Meanwhile, AdaProp [80] and

Table 9: Dataset Statistic. "LP" and "EC" denote link prediction and entity classification. "#" represents the number.

| Datasets | Tasks | #Entities | #Relationship | #Edges | #Train | #Validation | #Test |
|---|---|---|---|---|---|---|---|
| FB15K-237 | LP | 14,505 | 237 | 310,079 | 272,115 | 17,526 | 20,438 |
| WN18RR | LP | 40,559 | 11 | 92,583 | 86,835 | 2824 | 2924 |
| CoDEx-Large | LP | 77,951 | 69 | 612,437 | 551,193 | 30,622 | 30,622 |
| YAGO 3-10 | LP | 123,143 | 37 | 1,089,000 | 1,079,040 | 4978 | 4,982 |
| OGB WikiKG 2 | LP | 2,500,604 | 535 | 17,137,181 | 16,109,182 | 429,456 | 598,543 |
| WD50K | EC | 46,164 | 526 | 222,563 | 4600(N) | 4600(N) | 4600(N) |

One-shot subgraph method [81] propose novel subgraph-based efficient methods, which are also an emerging direction these years.

In this work, we focus more on developing a better and characteristic-adaptive anchor-based sampling strategy. Unlike other independent parameter-efficient methods, although the anchor-based methods may cause performance deduction, their plug-and-play attributes are better than other parameter-efficient methods. It has excellent potential to be easily integrated with any other KGE models, including hyperbolic embedding methods, such as ATTN [9]. Besides, among different types of efficiency of KGE models, such as parameter efficiency, time efficiency, and memory efficiency, we are concentrating more on the parameter efficiency of traditional KGE models instead of targeting time and memory complexity.

**Anchor-based Strategy.** Anchor-based strategies, as effective sampling techniques, are widely used in different fields [52, 73, 74, 67, 7] to solve efficiency problems these years. The main idea of these strategies is to select the representative sample to represent a group of data so that the whole dataset is not required for the learning procedure, which will improve the scalability and efficiency of the original model. Inspired by Subword-powered algorithms [4, 58] in NLP, NodePiece [20] is proposed, which is the most representative anchor-based KGE model. Many KGE models come out based on it, such as TripleRE [78]. Besides, StarGraph [30] and DigPiece [70]are proposed by adding extra neighbour constraints. Although these models all achieve promising performance, the anchors in these methods are still selected in a primitive manner, *e.g.,* random selection, and manual selection of different centrality measurement strategies, including Node Degree Centrality, Personalized PageRank, etc. Since these selection strategies are neither reasonable nor developed according to the characteristics of the data type of KGs, the quality of the anchors cannot be well guaranteed, which limits the performance of the models. In particular, the weight for each strategy on different datasets is determined according to grid searching, which is time-consuming.

### A.3   Experiment Setting

Experiment settings are introduced from three aspects, *i.e.,* datasets, implementation details, and evaluation metrics.

#### A.3.1   Datasets

Six benchmark datasets are leveraged to evaluate our RecPiece as same as previous works do [13, 20, 30]. Specifically, FB15k-237 [64], WN18RR [18], CoDEx-L [54], and YAGO3-10 [45] are used for link prediction. Among them, FB15k-237 [64] contains 237 relations, which is derived from Freebase [5], and inverse relations are deleted to avoid the leaking problem compared to FB15k. Similar to it, WN18RR [18] is derived from WordNet [47] without inverse relations. FB15k-237 can be considered relation-rich graphs, while WN18RR is a sparse graph with few relation types. Besides, YAGO3-10 [45] consists of entities that have a minimum of 10 relations each which are extracted from YAGO3. Different from the above three typical traditional datasets, CoDEx [54] contains more diverse and interpretable content and is more difficult to evaluate. In this work, we used the largest subset of it. The entity classification is carried out on two subsets (5% and 10% labeled) from WD50K [21]. Furthermore, we leverage OGB WIKIKG 2 [23, 20], a large-scale KG, for scalability analysis. The statistical details of the datasets are shown in Tab. 9.

#### A.3.2   Implementation Details

All experiments are conducted on the server with 4-core Intel(R) Xeon(R) Platinum 8358 CPUs @ 2.60GHZ, a single 80 GB A100 GPU, and 64GB RAM with PyTorch [49] libraries. The $p(\cdot)$

for feature preparation is selected as pretrained NodePiece [63] in the first few epochs. Besides, k-means [41, 44] is selected as $g(\cdot)$ for clustering, and the cluster number is set as "#Rel." in Tab. 9 for different datasets. Besides, a 2-layer-MLP is adopted as $f(\cdot)$ feature propagation. In addition, we replace the default $p(\cdot)$, $g(\cdot)$ and $\mathrm{dist}(\cdot)$ to pretrained GraIL [63], BitechingKmeans [50] and euclidean$(\cdot)$ for ablation analysis.

Moreover, as for different tasks, RecPiece is integrated with three KGE backbones, *i.e.,* RotatE [61], COMPGCN [68], and AutoSF [79] to compare with thirteen KGE models, including (1) link prediction: TransE [6], DisMult [77], ComplEX [65], PairRE [10], RotatE [61], TripleRE [78], AutoSF [79], LRE + PairRE [11], NodePiece + RotatE [20], and NodePiece + AutoSF [20]; (2) entity classification: MLP, COMPGCN [68], and NodePiece + COMPGCN [20]. Note that the anchor selection in our RecPiece is a relational clustering-based anchor selection strategy. Compared to the manual selection strategy (composed of 20% Random Selection, 40% Personalized PageRank, and 40% Node Degree Centrality) in NodePiece, ours is more explainable and scalable. Here we present some descriptions of the used models as follows.

- RotatE defines each relation as a rotation from the source entity to the target entity in the complex vector space, enabling it to model and infer various relation patterns, including symmetry, antisymmetry, inversion, and composition relations.
- COMPGCN encodes entities and relations jointly by using various composition operators from KGE techniques, addressing the issue of over-parameterization in GCNs.
- AutoSF proposed an algorithm that can automatically design and discover optimal scoring functions of the KGE model. Through a progressive greedy search algorithm, AutoSF can design promising KGE scoring functions effectively from a vast search space.
- TransE is a translation-based KGE model that aims to model inversion and composition relations. Inspired by the translation invariance in the word2vec model, TransE tries to make h + r≈t, where h, r, and t represent the head entity, relation, and tail entity in a triplet, respectively.
- DistMult assumes that all relations in the KG are symmetric and represent them as block-diagonal matrices. Such a relation representation mechanism, combined with simple dot product operations, improves the efficiency of triplet evaluation.
- ComplEx uses complex vectors instead of real vectors to represent the embeddings of entities and relations, which allows the model to distinguish between symmetric and asymmetric relations.
- PairRE uses paired vectors to represent each relation, allowing the margin in the loss function to adjust adaptively. Thus, PairRE can express more complex relations, such as sub-relations.
- TripleRE combines projection and translation operations. Specifically, the representation vectors of the head and tail entities are first projected and then translated to obtain the relation representation. This method enriches the expression of relations, enabling the model to handle complex relations.
- NodePiece proposed an efficient and plug-and-play node selection mechanism for KGE models. Specifically, NodePiece is inspired by WordPiece from the field of natural language processing, which is able to represent large-scale knowledge graphs using only fewer entity embeddings while also enhancing the generalization performance of the model at the same time.
- LRE is a high-efficiency method that utilizes tensor decomposition to enhance the parameter efficiency of KGE models. Specifically, rather than decomposing the observed 3D tensor directly, LRE decomposes the entity embedding matrix to low-rank matrices.

### A.3.3 Evaluation Metrics

Two types of evaluation metrics are adopted for two different downstream tasks. For link prediction, both mean reciprocal rank (MRR) [16] and Hits@k [1] are used as the ranking metrics, where k $\in$ {1, 3, 10}. Besides, ROC-AUC, PRC-AUC, AP, and Hard Accuracy are the evaluation metrics [23] for entity classification. To quantify the efficiency, we report the parameter number #P (M), memory cost (GB), running time (hours), and *Effi.* [13], which is the metric recently proposed by [13] to evaluate the efficiency of the KGE models.

$$Effi. = \frac{\mathrm{MRR}}{\#\,\mathrm{Parameter}} \tag{8}$$

Note that higher *Effi.* it is, the more efficient the model performs.

