# OpenReview forum: "Clustering then Propagation: Select Better Anchors for Knowledge Graph Embedding"
_NeurIPS.cc/2024/Conference — NeurIPS 2024 poster_

### Official Review · Reviewer_vJzB · 2024-07-01

**Soundness:** 3
**Presentation:** 3
**Contribution:** 3
**Rating:** 6
**Confidence:** 4

**Summary:**

The paper presents an interesting idea to make the existing KGE methods achieve better efficiency, termed RecPiece. It is proposed based on two characteristics, i.e., representative ability of cluster centroids and the descriptive ability of the relational facts. Complete experiments are carried out based on five datasets, two downstream tasks, and three KGE backbones, which shows its promising performances.

**Strengths:**

1.	The paper is overall easy to follow, and readable.
2.	The idea of the paper is interesting and practical. It makes use of the clustering strategy to generate the representative data first, and then, performs propagation based on it. Although such prototype ideas are appeared in other fields, like computer vision, etc. From my experiences, it is first applied for KGE.
3.	The authors explain why they select relational facts for clustering, instead of directly using entity features for clustering. I really like the thoughts behind it that the authors carefully think about the differences between KGs and other general graphs, and take use of it.
4.	The experiments are complete. As I mentioned in summary above, various experiments are carried out based on five datasets over two downstream tasks (4 for link prediction and 1 for entity classification). Besides, three KGE backbones, RotatE, CompGCN, and AutoSF, are leveraged. Moreover, experiments demonstrate the performances of their model from 6 aspects.

**Weaknesses:**

You should explain more about the compared baselines in your experiment settings, even with one sentence. It can make the reader better understand your paper.

**Questions:**

Is it possible to extend your idea in inductive settings?

I would recommend the authors discuss more about making use of multi-modal attributes for feature preparation in the future work.

---

> ### Author Rebuttal · Authors · 2024-08-06
>
> Thanks for your valuable suggestions! We respond to your questions carefully one by one carefully, and we hope our responses can address your concerns.
>
> **RQ1. More Description on KGE Baselines, and Evaluation Metrics.**
>
> Thanks for your suggestions. I will add the description of the baselines in our final version of the manuscript. As for the evaluation metrics, we have already introduced them in Appendix A.3.3. We will also add more descriptions of them in our final version. Concretely, the descriptions of the compared KGE baselines are presented as follows:
>
> **RotatE** defines each relation as a rotation from the source entity to the target entity in the complex vector space, enabling it to model and infer various relation patterns, including symmetry, antisymmetry, inversion, and composition relations.
>
> **COMPGCN** encodes entities and relations jointly by using various composition operators from KGE techniques, addressing the issue of over-parameterization in GCNs.
>
> **AutoSF** proposed an algorithm that can automatically design and discover optimal scoring functions of the KGE model. Through a progressive greedy search algorithm, AutoSF is able to design promising KGE scoring functions effectively from a vast search space.
>
> **TransE** is a translation-based KGE model that aims to model inversion and composition relations. Inspired by the translation invariance in the word2vec model, TransE tries to make h + r≈t, where h, r, and t represent the head entity, relation, and tail entity in a triplet, respectively.
>
> **DistMult** assumes that all relations in the KG are symmetric and represent them as block-diagonal matrices. Such relation representation mechanism, combined with simple dot product operations improves the efficiency of triplet evaluation.
>
> **ComplEx** uses complex vectors instead of real vectors to represent the embeddings of entities and relations, which allows the model to distinguish between symmetric and asymmetric relations.
>
> **PairRE** uses paired vectors to represent each relation, allowing the margin in the loss function to adjust adaptively. Thus, PairRE can express more complex relations, such as sub-relations.
>
> **TripleRE** combines projection and translation operations. Specifically, the representation vectors of the head and tail entities are first projected and then translated to obtain the relation representation. This method enriches the expression of relations, enabling the model to handle complex relations.
>
> **NodePiece** proposed a efficient and plug-and-play node selection mechanism for KGE models. Specifically, NodePiece is inspired by WordPiece from the field of natural language processing, which are able to represent large-scale knowledge graphs using only fewer entity embeddings, while also enhancing the generalization performance of the model at the same time.
>
> **LRE** is a high-efficiency method that utilizes tensor decomposition to enhance the parameter efficiency of KGE models. Specifically, rather than decomposing the observed 3D tensor directly, LRE decomposes the entity embedding matrix to low-rank matrices.
>
>
> **RQ2. Inductive Settings**
>
> Thanks for your question. The current version of our RecPiece cannot handle the inductive settings, as the clustering-based anchor selection strategy is performed on the preprocessed features of the whole graphs. However, unseen entities or relations exist in inductive settings, where will lead the bias for anchor selection during training. But thanks for your valuable comments, we will try to work on it in the future.
>
> **RQ3. Utilization of Multi-modal Attributes**
>
> Thanks for your valuable advice, and we will answer your questions as follows.
>
> **(1)**	Sure, it is definitely possible to utilize other multi-modal attributes, and we have already considered it. Specifically, the performance comparison and discussion are present between different prepared features from both structural information and textual information in Sec. 4.2.2 and Tab. 4 of our paper.
>
> **(2)**	We further prepare the clustering features from visual information. Specifically, we make use of the multi-modal version of the FB15k-237 datasets from [1]. Then, we further leverage ViT [2] to generate the visual features of each entity. According to the Tab.1 in the attached PDF, we can also see that our RecPiece can achieve better performance with the visual attributes, which further demonstrates the scalability and effectiveness of our method.
>
>
>
>
> **References.**
>
> [1] A survey of knowledge graph reasoning on graph types: Static, dynamic, and multi-modal
>
> [2] An Image is Worth 16x16 Words: Transformers for Image Recognition at Scale.

---

### Official Review · Reviewer_89jh · 2024-07-01

**Soundness:** 3
**Presentation:** 3
**Contribution:** 3
**Rating:** 7
**Confidence:** 3

**Summary:**

A clustering-guided anchor-based efficient KGE method is proposed in the paper. Concretely, it takes advantages of the clustering, which can be treated as an effective sampling way compared to random selected. Based on such idea, the authors apply the mechanism to different backbones and different downstream tasks. The experimental results show their great performances compared to previous anchor-based KGE method, i.e., NodePiece.

**Strengths:**

1. The idea is first seen in knowledge graph representation learning from my aspect. It is novel, which is also simple and proven effective as shown in experiments.

2. The experiments are sufficient, which shows the performances of the model from six aspects. Besides, three KGE models are adopted as the basic backbones for comparison. Moreover, two downstream tasks are evaluated. All these experiments sufficiently support the main claim of the proposed model.

3. The paper takes attempt to address an important task in KGE, but with a simple yet effective strategy which can be easily applied to other GNN based models, especially in real-world scenarios.

**Weaknesses:**

1. Authors should double check the paper for writing mistakes, such as Superiority in line 191, etc.

2. As LLM rapid developed, I am curious about whether we can integrate LLM-based models to enhance your strategy.

3. Experimental settings should be described more, such as description of compared baselines, explanation of evaluation metrics, etc.

**Questions:**

1. Although the clustering preparation is beneficial for anchor selection, it will also bring additional computational overhead. How can we optimize such overhead in the future?

2. Can you explain the advantages more about relational clustering-based anchor selection strategy in the proposed model?

---

> ### Author Rebuttal · Authors · 2024-08-06
>
> Thanks for your valuable suggestions! We respond to your questions carefully one by one carefully, and we hope our responses can address your concerns.
>
> **RQ1. Writing Issues.**
>
> Thanks for your suggestions. I will reorganize our paper into shorter paragraphs, especially for the second and third paragraphs of the introduction and related works. We will also double-check our paper to revise typo mistakes, reference redundancy and unify the reference formats.
>
> **RQ2. Corporation with our RecPiece and LLMs.**
>
> Thanks, and we will respond from from two aspects.
>
> **(1) Enhancing RecPiece with LLMs.** As shown in Fig. 1 of our manuscript, there are four steps in our RecPiece. As the main goal for our RecPiece is to achieve a better trade-off between efficiency and effectiveness, integrating LLMs into our framework, especially for the embedding models, will raise the parameter redundancy except for the first step, i.e., feature preparation. Concretely, there are two potential ways to utilize LLMs. On the one hand, we can utilize LLMs for textual information encoding, as shown in Fig. 1 (a) in the attached PDF. Actually, we have already tried to leverage PLMs for it in Sec. 4.2.2 and Tab. 4. Compared to PLMs, LLMs are of better expressive capacity, which will lead to better representation. On the other hand, when applying our RecPiece to real-world scenarios, we can use LLM-based agents for real-time information retrieval and collection, as shown in Fig. 1 (b) in the attached PDF. It saves the expensive labor cost for information collection before feature preparation.
>
> **(2) Enhancing LLMs with RecPiece.** Our RecPiece can be treated as an efficient feature encoding method, which is an option for utilizing large-scale KGs in our real-world applications. With the generated embeddings for the structural knowledge, LLMs can be more expressive for different downstream tasks [1] (As shown in Fig. 1 (c) in the attached PDF).
>
> **RQ3. More Description on KGE Baselines, and Evaluation Metrics.**
>
> Thanks for your suggestions. I will add the description of the baselines in our final version of the manuscript. As for the evaluation metrics, we have already introduced them in Appendix A.3.3. We will also add more descriptions of them in our final version. More details can be checked in RQ1 of Reviewer vJzB.
>
> **RQ4. Optimization on Redundant Overhead of Clustering-based Strategy**
>
> Thanks for your valuable advice. The question you mention is a common one for all anchor-based KGE models, where the anchor selection procedure is inevitable yet crucial. In the future, we can integrate the anchor selection procedure with the model learning procedure. Besides, we only need to run the clustering-based anchor selection algorithm once. With better-quality anchors, better performances can be achieved. From this view, the redundant overhead for anchor selection is also acceptable.
>
> **RQ5. Advantages of our Relational Clustering-based Anchor Selection Strategy**
>
> Thanks and we will answer your questions from three aspects, which are actually described in Sec. 3.6 of our manuscript.
>
> **(1)** Random and manual anchor selection in previous anchor-based models is highly dependent on the human experience. Compared to them, ours is more reasonable and learnable according to the representative ability of the cluster centroids.
>
> **(2)** Our RecPiece is developed based on the characteristics of KGs. Specifically, we perform clustering on features of triplets instead of entities since the knowledge units in KG are stored in triplets, which can also be easily characterized based on the relation type.
>
> **(3)** The hyper-parameter, i.e., cluster number, for clustering algorithms can be determined according to the attributes in KGs in RecPiece as the number of relation types. Thus, our anchor selection only contains one hyper-parameter, i.e., anchor number, which is inevitable and also needed by other anchor-based methods. Besides, other models even need resource-consuming grid-searching to get weights for different centrality measurement strategies on different KGs.
>
> **Reference**
>
> [1] Unifying large language models and knowledge graphs: A roadmap.

---

> > ### Comment · Reviewer_89jh · 2024-08-11
> >
> > Thank you for your responses, which have addressed my concerns.

---

### Official Review · Reviewer_GCgr · 2024-07-07

**Soundness:** 3
**Presentation:** 3
**Contribution:** 3
**Rating:** 7
**Confidence:** 4

**Summary:**

To address the computational inefficiencies of conventional knowledge graph embedding models, the authors of this study suggest RecPiece, an anchor selection technique based on relational clustering. RecPiece selects more efficient anchor entities by using the descriptive power of relation types and the representative ability of cluster centroids, which improves efficiency and scalability over earlier anchor-based methods.

**Strengths:**

The overall writing of the paper is clear and well-structured. Clear figures are provided to help understand the proposed method, and the notations used are generally clear.

The efficiency and scalability problem investigated is important to the graph community. The paper is about a practical problem for graph-based scenarios.

The proposed method is simple yet effective. Specifically, it is a novel anchor-based method for knowledge graph representation that leverages clustering centroids and relation types. Compared to primitive anchor selection strategies, there is a great improvement as RecPiece can provide more representative and descriptive anchors for better performance on six datasets.

The experiments cover several datasets, with link prediction tasks, entity classification tasks, and relation prediction tasks. The scope of the experiments is extensive. Several ablation studies and empirical analyses are provided.

**Weaknesses:**

Only one embedding model, RotatE, is considered in experiments on datasets FB15k-237, WN18RR, CoDEx-L, and YAGO 3-10 (see Table 1). Other embedding models, such as TransE, DistMult, and ComplEx, should also be included here.

The writing of the paper can be improved. Some long paragraphs can be split into shorter ones. Besides, there are some repeated references, such as [1] and [2], [11] and [12], [51] and [52]. It would be better to check these references and unify the formats.

**Questions:**

Several GNN methods work on the efficiency and scalability problem of KGs, such as AdaProp (KDD'23), AStarNet (NeurIPS'23), and One-shot-subgraph (ICLR'24). I suggest the paper discuss these recent works.

As large language models (LLMs) are widely studied, I recommend that the author discuss more about using LLMs for efficient KGE in future works.

Please also refer to the above weaknesses.

**Limitations:**

NA.

---

> ### Author Rebuttal · Authors · 2024-08-06
>
> Thanks for your valuable suggestions! We respond to your questions carefully one by one carefully, and we hope our responses can address your concerns.
>
> **RQ1. More Compared KGE Backbones.**
>
> Thanks for your suggestions, and we will answer your questions from three aspects as follows.
>
> **(1)**	Actually, RecPiece utilizes three kinds of KGE backbones instead of one model, including RotatE, CompGCN, and AutoSF, which reviewers vJzB and 89jh also mentioned.
>
> **(2)**	For a fair comparison, we follow the experiment settings in NodePiece [1]. Different KGE backbones are adopted for different downstream tasks. For example, RotatE is the KGE backbone for performance evaluation on link prediction on normal KG benchmarks and we also integrate our RecPiece with only one KGE backbone, i.e., RotatE, as same as NodePiece.
>
> **(3)**	As for TransE, DistMult and ComplEx, they are actually compared in the scalability analysis of Sec. 4.4 and Tab. 6 of the manuscript (Refer Tab. 2 in the attached PDF).
>
> **RQ2. Writing Issues**
>
> Thanks for your suggestions. I will reorganize our paper into shorter paragraphs, especially for the second and third paragraphs of the introduction and related works. Besides, we will double-check our paper to revise typo mistakes, reference redundancy and unify the reference formats.
>
> **RQ3. Discussion on Advanced Models**
>
> Thanks for your suggestions. Those are indeed advanced GNN methods that work on the efficiency and scalability problem of KGs, and we will discuss them in our revised version. Concretely, the discussion on them, including AdaProp [2], AStarNet [3], and One-shot-subgraph [4], will be presented in the second part of the related work section, i.e., Parameter-Efficient Model, because they are not anchor-based KGE models.
>
> **RQ4. Corporation with our RecPiece and LLMs**
>
> Thanks for your valuable suggestions. We will answer it from two aspects, and add the discussion in our revised paper as the future works.
>
> **(1) Enhancing RecPiece with LLMs.** As shown in Fig. 1 of our manuscript, there are four steps in our RecPiece. As the main goal for our RecPiece is to achieve a better trade-off between efficiency and effectiveness, integrating LLMs into our framework, especially for the embedding models, will raise the parameter redundancy except for the first step, i.e., feature preparation. Concretely, there are two potential ways to utilize LLMs. On the one hand, we can utilize LLMs for textual information encoding, as shown in Fig. 1 (a) in the attached PDF. Actually, we have already tried to leverage PLMs for it in Sec. 4.2.2 and Tab. 4. Compared to PLMs, LLMs are of better expressive capacity, which will lead to better representation. On the other hand, when applying our RecPiece to real-world scenarios, we can use LLM-based agents for real-time information retrieval and collection, as shown in Fig. 1 (b) in the attached PDF. It saves the expensive labor cost for information collection before feature preparation.
>
> **(2) Enhancing LLMs with RecPiece.** Our RecPiece can be treated as an efficient feature encoding method, which is an option for utilizing large-scale KGs in our real-world applications. With the generated embeddings for the structural knowledge, LLMs can be more expressive for different downstream tasks [2] (As shown in Fig. 1 (c) in the attached PDF).
>
>
> **References.**
>
> [1] NodePiece: Compositional and Parameter-Efficient Representations of Large Knowledge Graphs.
>
> [2] Adaprop: Learning adaptive propagation for graph neural network based knowledge graph reasoning.
>
> [3] A* net: A scalable path-based reasoning approach for knowledge graphs.
>
> [4] Less is More: One-shot Subgraph Reasoning on Large-scale Knowledge Graphs.
>
> [5] Unifying large language models and knowledge graphs: A roadmap.

---

> > ### Comment · Reviewer_GCgr · 2024-08-12
> >
> > Thanks for the detailed responses and additional experiments, which solve my concerns and questions. I will raise my rating.

---

### Official Review · Reviewer_z3Fe · 2024-07-08

**Soundness:** 3
**Presentation:** 3
**Contribution:** 3
**Rating:** 6
**Confidence:** 4

**Summary:**

This paper proposes RecPiece, a novel anchor-based knowledge graph embedding (KGE) model that selects representative anchors via a relational clustering-based strategy. Specifically, RecPiece performs clustering over features of factual triplets instead of entities to generate cluster centroids, and the cluster number is set as the number of relation types. Representative triplets are then selected around the centroids and mapped to corresponding anchor entities. Extensive experiments on link prediction and entity classification show RecPiece achieves better performance with comparable or fewer parameters than previous anchor-based KGE models like NodePiece, demonstrating its ability to select better anchors in a more scalable way.

**Strengths:**

1. The proposed relational clustering-based anchor selection strategy leverages the representative ability of cluster centroids and descriptive ability of relation types in knowledge graphs. This is a novel and reasonable approach compared to previous random or manual anchor selection.

2. RecPiece is developed based on the characteristics of knowledge graphs by clustering on triplet features and using the number of relation types as cluster number. This makes the method more suitable for the data type of knowledge graphs.

3. Extensive experiments on link prediction and entity classification across multiple datasets demonstrate the superiority, effectiveness, efficiency, scalability and transferability of RecPiece over baselines. For example, RecPiece achieves 3.5% and 2.5% improvements on MRR and Hits@10 for link prediction with 6.5% fewer parameters compared to NodePiece.

4. RecPiece provides a simple yet effective plug-and-play module that can be easily integrated with various KGE models to reduce their space complexity and improve efficiency without incurring significant performance loss.

**Weaknesses:**

1. While RecPiece shows promising results, it still has some performance loss compared to the non-anchor based KGE models on some datasets. Though such trade-off between efficiency and effectiveness is expected, it would be better to provide some theoretical analysis on the potential causes of the performance gap.

**Questions:**

1. How does the choice of the clustering algorithm affect RecPiece's performance? Will more advanced clustering methods bring further improvements?

2. Besides the structure information, is it possible to make use multi-modal information during feature preparation?

3. Is it possible to make use of LLM to enhance RecPiece in the future?

---

> ### Author Rebuttal · Authors · 2024-08-06
>
> Thanks for your valuable suggestions! We respond to your questions carefully one by one carefully, and we hope our responses can address your concerns.
>
> **RQ1. More Discussion on Trade-off between Efficiency and Effectiveness.**
>
> Thanks, and we will respond from two aspects.
>
> **(1)**	The anchor-based KGE methods aim to narrow the view scope from the entire graph to the critical entities, thereby improving the efficiency of the model. As less information is utilized via this strategy, performance loss is usually inevitable, which also occurs on our RecPiece, a typical anchor-based KGE model. However, compared to previous anchor-based models, e.g., NodePiece, RecPiece can reduce performance loss and improve efficiency. In other words, RecPiece can achieve a promising performance with a better trade-off between efficiency and effectiveness.
>
> **(2)**	Furthermore, we also agree with you and provide a theoretical analysis to illustrate two attributes, i.e., (i) the performance loss of the anchor-based KGE methods is inevitable, and (ii) the samples selected by RecPiece are more representative than random selection.
>
> **(i)** As mentioned in (1), anchor-based KGE methods actually try to select a proportion of the dataset to represent the whole dataset. However, the performance loss is inevitable with fewer samples, which is proven in many classic researches [1][2]. Here, we demonstrate it by relying on Theorem 1 in [1].
>
> $\textbf{Theorem 1.}$ Assume $0<\epsilon\leq\frac{1}{8}$, $0<\delta\leq\frac{1}{100}$ and the  $VCdim(C)\geq2$. The any $(\epsilon, \delta)$-learning algorithm A for C must use sample size
>
> $$m_A(\epsilon, \delta)\geq\frac{VCdim(C)-1}{32\epsilon}=\Omega(\frac{ VCdim(C)}{\epsilon}) (1) $$
>
> Note that the Vapnik-Chervonenkis dimension [3] of C is denoted as VCdim(C), which is the cardinality of the largest $W\subseteq C$ such that $W$ is shattered by $C$. The $\epsilon$ is the expected error.
>
> Based on the (1) above, we can get the following inequality (2)
>
> $$\epsilon \geq\frac{VCdim(C)-1}{32 m_A(\epsilon, \delta)} (2),$$
>
> where we can easily get that when $VCdim(C)\geq2$, the expected error $\epsilon$ is lower bounded by $\frac{VCdim(C)-1}{32m_A(\epsilon, \delta)}$. As the VCdim(C) in our case is determined as a constant once the learning algorithm $A$ is given, the lower bound of the expected error has negative correlations with the number of the samples $m_A$. Thus, fewer samples will lead to a larger lower bound, which will further result in worse performance.
>
> **(ii)** Assume the size of the original datasets is $N$, and the number of the categories is $k$, the more representative sampled dataset should at least follow one characteristic that there exists at least one sample belonging to each category. To demonstrate that our RecPiece makes it possible to select more representative samples compared to random selection, we can calculate the probability of both strategies for the abovementioned characteristics. Concretely, if the size of the sampled dataset is $m$ ($m\geq k$), there are $C_N^m$ combinations for random selection, and the probability for it is $\frac{1}{C_N^m}\ll 1=\operatorname{Prob(RecPiece)}$. It shows that the sampled sub-datasets generated from our strategy will contain all kinds of knowledge in the original datasets, which is more likely to lead to less performance loss.
>
> **RQ2. Influences of Clustering Algorithms.**
>
> Thanks, and we will respond from three aspects.
>
> **(1)** The performance of anchor-based KGE methods highly corresponds to the quality of the anchors. Inspired by the representative capacity of the clustering centroid, our RecPiece utilizes clustering for anchor selection. If different clustering algorithms can all find the same or similar clustering centroids, the clustering algorithms will have less effect on RecPiece performance as the selected anchors may also be similar. Otherwise, different clustering algorithms will affect the RecPiece’s performances by affecting the selection of the anchor sets.
>
> **(2)** In our work, we focus more on introducing the idea of leveraging clustering for anchor selections in anchor-based KGE methods and emphasize the importance and advantages of performing clustering on relational facts rather than entities. Therefore, we use the most commonly used but effective clustering algorithm, $k-$means, to evaluate our idea.
>
> **(3)** Moreover, we discuss the influences of different clustering algorithms in Sec. 4.2.3 and Fig. 2 (b). Our paper provides more details.
>
> **RQ3. Utilization of Multi-modal Attributes.**
>
> Thanks, and we will respond from two aspects.
>
> **(1)**	Sure, it is definitely possible to utilize other multi-modal attributes, and we have already considered it. Specifically, the performance comparison and discussion are present between different prepared features from both structural information and textual information in Sec. 4.2.2 and Tab. 4 of our paper.
>
> **(2)**	We further prepare the clustering features from visual information. Specifically, we make use of the multi-modal version of the FB15k-237 datasets from [4]. Then, we further leverage ViT [5] to generate the visual features of each entity. According to the Tab.1 in the attached PDF, we can also see that our RecPiece can achieve better performance with the visual attributes, which further demonstrates the scalability and effectiveness of our method.
>
>
> **RQ4. Corporation with our RecPiece and LLMs.**
>
> Thanks! Due to the space limitation, please refer to RQ4 for Reviewer GCgr.
>
> **References.**
>
> [1] A general lower bound on the number of examples needed for learning
>
> [2] An introduction to computational learning theory
>
> [3] Learnability and the Vapnik-Chervonenkis dimension
>
> [4] A survey of knowledge graph reasoning on graph types: Static, dynamic, and multi-modal
>
> [5] An Image is Worth 16x16 Words: Transformers for Image Recognition at Scale.

---

> > ### Comment · Reviewer_z3Fe · 2024-08-11
> >
> > Thanks for the detailed responses. My concerns are addressed. Besides, I have also checked the comments of other reviewers. Overall, the paper is of good quality. Thus I prefer to raise my score to 6.

---

### Author Rebuttal · Authors · 2024-08-06

We thank the SAC, AC, and PCs for their efforts and constructive comments, which are helpful in further improving the quality of our manuscript. We respond to your questions carefully one by one carefully, and we hope our responses can address your concerns.

Note that there are two tables and one figure in the attached PDF, corresponding to RQ3 and RQ4 for Reviewer z3Fe, RQ1 and RQ4 for Reviewer GCgr, RQ2 for Reviewer 89jh, and RQ3 for Reviewer vJzB.

---

### Decision · Program_Chairs · 2024-09-25

**Decision:**

Accept (poster)

**Comment:**

This submission proposes a modification to NodePiece [18] by changing the clustering scheme and encodings shown in eqns. (1) and (2) in [18] to incorporate a relation representation that is based on the (head, tail) pairs connected by the relatio.  This submission is not self-contained; I eventually had to go and read [18] (a more creative and much better-written paper) to make sense of the current submission.  The modification made to [18] feels relatively minor.  The (head, tail) based representation of relations has been used for a few years (see citation below).  There is some improvement in KG completion (0.3-5.2 relative percent) and entity classification (0.3%), which are not earth shattering.


Beyond numeric scores, this submission sinks or swims based on its marginal contributions beyond [18] (NodePiece).  Table 1 has a succinct summary.  Observe that model size savings from RotatE to NodePiece are 10x for FB15k237, 8x for WN18RR, 20x for CoDEx-L, 30x for YAGO3-10.  In comparison, the model size savings from NodePiece to RecPiece are 9% for FB15k237, 0% for WN18RR, 17% for CoDEx-L and 4% for YAGO3-10.  Similar observations hold for Table 2.  These are orders of magnitude apart, so most of the heavy lifting was done by the key innovation in NodePiece.

Accuracy improvements reported in Tables 3 and 4 are also very modest.

In summary, it is possible to argue that both the technique and benefit beyond NodePiece are modest, except for the OGB WikiKG dataset, which was likely the authors' primary target.

If the panel considers the incremental progress still important to publish and report, they should also consider the fact that the manuscript is rather poorly written.


Only that last part is fixable. Some suggestions follow.

L41 This is the first occurrence of the word "propagation" after the abstract.  The typical reader might benefit from a 1-2 sentence disclosure of what "propagation" means in this context.  What exactly are you propagating "from an intact set of all of the entities to an anchor set of some representative entities"?

To keep the manuscript self-complete and not require the reader to first read [18], "anchor based sampling" should be explicated further in the intro itself.  E.g. up to the end of page 2, we do not know if an anchor is an entity, or an aggregation over entities, or a relation type, or a representative relation type.  Unfortunately, without the assistance of concrete notation, the intro fails to give anything beyond a remote hint of the technique.

The caption (not distant text) of Figure 1 should include these pieces of info:
* Before entering $p(\cdot)$, what are node representations?  Do they depend on any textual aliases of nodes?
* Is the feature preparation capable of gradient-based updates, or is a fixed feature computation function?
* Are the colored vectors between the applications of $p(\cdot)$ and $g(\cdot)$ mutable through the training process or fixed?
* In the topmost row, $B$ get a shade of blue, but in the third row, it gets an orange shade.  What is the significance of this?
* If $|\mathcal{R}|$ is the number of distinct relation types, it seems there is one point cloud for each relation type, characterized by the features of the entity-pairs participating in that relation type.
* After application of $\phi _a$ we see cluster centroids marked.
* After application of $\phi _b$ we see some nodes in the original graph acquire a color, but some other nodes do not.  Explain why.
* How is $f(\cdot)$ a _propagation_?  Around this point the diagram gets too murky to follow.

L109 Link prediction in the context of KGs is just KG inference or completion, but what is node (entity) classification?  What are labels?

L117 "Thus, we selected the features of triplets as the clustering features in this paper." ― what _are_ the features of the triplets?

L122 and we still have no idea what "feature propagation" means.

Between equations (1) and (2) there are various sources of confusion.

L132 "first generate the embeddings of entities and relations" ― how?  Must we inspect your code to find out this critical detail?

Nit: If you will be using $h$ for embedding, do not use $h$ for head, use subject and object instead.

L133-L134 is equally problematic. What is a "normalized embedding"?  Assuming $\bar{h} _{\text{head}}$ and $\bar{h} _{\text{tail}}$ have the same width, and normalization does not change the length of a vector, how do you "sum" these with $\bar{h} _r$?

I know what you are doing here, just that the notation is a disaster.

On L134, why isn't $n$ also attached to $e _h$ and $e _t$?

On L135 you are about to define $\mathbf{H} _t$ but then suddenly switch to $\mathbf{H} _\mathcal{G}$ in the lhs in eqn. (2).  Is this a typo?

In eqn. (2), $\bigoplus$ means concatenation.  Depending on the popularity of the relation type involved, you therefore get representations of variable lengths?

Using multisets of (subject, object) pair embeddings to represent a relation has been known in the KG literature for some time:
https://ieeexplore.ieee.org/document/10191597  (see Eqn (4))

In eqn (3), is there any reason to set the number of clusters to precisely $|\mathcal{R}|$?

L151 Nit: Cosine is a similarity, not a distance.

L157 So at this point, for every relation type $i \in [ | \mathcal{R} | ]$, you collect $m _i$ triples, for a total of $\sum _i m _i$ exemplar triples?  Why possible different $m _i$s for different relation types (eq. 6)?

L166 What _are_ "the features of each entity $e _i$"?  The reader should not wait for section 4.2.2.

In Section 3.5 of your submission, you mention propagation in the context of [18] but the NodePiece paper has zero occurrence of the word "propagation".